# Effects and Mechanisms of Cage versus Floor Rearing System on Goose Growth Performance and Immune Status

**DOI:** 10.3390/ani13162682

**Published:** 2023-08-21

**Authors:** Bincheng Tang, Shenqiang Hu, Xin Zhang, Qingyuan Ouyang, Enhua Qing, Wanxia Wang, Jiwei Hu, Liang Li, Jiwen Wang

**Affiliations:** 1Farm Animal Genetic Resources Exploration and Innovation Key Laboratory of Sichuan Province, Sichuan Agricultural University, Chengdu 611130, China; i_cbtang@163.com (B.T.); shenqianghu@gmail.com (S.H.); yibo08051004@163.com (X.Z.); oyqy222@163.com (Q.O.); qingenhua@yeah.net (E.Q.); hujiwei1990@126.com (J.H.); liliang@sicau.edu.cn (L.L.); 2General Station of Animal Husbandry of Sichuan Province, Chengdu 610066, China; 18225042288@163.com

**Keywords:** Sichuan white goose, rearing system, growth performance, immune status, transcriptomic analysis

## Abstract

**Simple Summary:**

The rearing system is one of the significant non-genetic factors that influence growth performance and immune status in goose husbandry. Therefore, it is of great significance to understand the effects of two major dryland systems on goose growth performance and immune status. The current study revealed that the floor rearing system (FRS) improved growth performance, but the immune function of geese was promoted to some extent in the cage-rearing system (CRS). Moreover, we found the phenylalanine metabolism pathway could exert positive effects on the immune function of geese under CRS. These findings can provide a reliable reference for selecting a dryland rearing system in geese.

**Abstract:**

Currently, FRS and CRS are the two predominant dryland rearing systems in the goose industry. However, the effects of these two systems on goose growth performance and health, as well as the underlying mechanisms, have not been fully clarified. Thus, this study aimed to compare growth performance and immune status, as well as investigate the genome-wide transcriptomic profiles of spleen in geese, between CRS and FRS at 270 d of age. Phenotypically, the body weight and body size traits were higher in geese under FRS, while the weight and organ index of spleen were higher in geese under CRS (*p* < 0.05). Noticeably, the bursa of Fabricius of geese under FRS was degenerated, while that under CRS was retained. At the serum level, the immune globulin-G (IgG) and interleukin-6 (IL-6) levels were higher in geese under CRS (*p* < 0.05). At the transcriptomic level, we identified 251 differentially expressed genes (DEGs) in the spleen between CRS and FRS, which were mainly enriched in scavenger receptor activity, inflammatory response, immune response, neuroactive ligand–receptor interaction, phenylalanine metabolism, ECM receptor interaction, calcium signaling pathway, phenylalanine, tyrosine, and tryptophan biosynthesis, regulation of actin cytoskeleton, and MAPK signaling pathways. Furthermore, through protein–protein interaction (PPI) network analysis, ten candidate genes were identified, namely, *VEGFA*, *FGF2*, *NGF*, *GPC1*, *NKX2-5*, *FGFR1*, *FGF1*, *MEIS1*, *CD36*, and *PAH*. Further analysis demonstrated that geese in CRS could improve their immune ability through the “phenylalanine metabolism” pathway. Our results revealed that the FRS improved growth performance, whereas the CRS improved goose immune function by increasing levels of IL-6 and IgG in serum. Moreover, the phenylalanine metabolism pathway could exert positive effects on immune function of geese under CRS. These results can provide reliable references for understanding how floor and cage rearing systems affect goose growth performance and immune capacity.

## 1. Introduction

With the rapid development of goose husbandry, the market demands for goose products are increasing. Meanwhile, as increasing attention has been focused on environmental protection and production efficiency, the rearing method of geese is transitioning from free range with water to modern dryland systems [1]. Currently, the net rearing system (NRS), cage rearing system (CRS), and floor rearing system (FRS) are the most common dryland rearing models in China [2], which greatly minimize the need for water bodies [3]. Previous studies have indicated that the rearing systems can affect poultry comfort, welfare, health status, intestinal microbial composition, and production performance [1,4,5,6]. However, in many countries of the world, the cage rearing system in geese is unknown. Geese are reared in an extensive or semi-intensive system (with access to a paddock and water (swimming pools, ponds, lakes). Thus, it is crucial to explore the growth performance and health status of geese under different rearing systems.

Recently, several studies have investigated the production performance and health status of poultry under different rearing systems. Zhang et al. [7] found that, compared to FRS, ducks under NRS showed a higher final body weight (BW), average daily gain (ADG), and average daily feed intake (ADFI), but a lower feed conversion rate (FCR), and there have also been significant changes in biochemical indicators of duck serum between FRS and NRS. In goose, Li et al. [1] showed that CRS significantly changes intestinal morphological characteristics and microbial composition, thereby affecting goose physiological functions and slaughter traits. Furthermore, in a study by Zhao et al. [8], the Shaoxing ducks reared in NRS had better health status compared to traditional rearing. It has been observed that the FRS appeared to be propitious for gastrointestinal development and health of broilers [9]. Moreover, NRS can improve duck immune function by promoting the development of major immune organs and increasing serum immunity cytokines levels [10]. However, CRS as a novel rearing system for geese, and the influence of growth performance and health status remains unclear.

Therefore, this experiment aimed to (1) explore the effects of different rearing systems (CRS vs. FRS) on goose growth performance and immune status, and (2) compare the spleen transcriptome profiles between CRS and FRS using RNA-seq. These results can help further understand how floor and cage rearing systems affect goose growth performance and immune capacity.

## 2. Materials and Method

### 2.1. Ethics Statement

In this study, all Sichuan white geese (SWG) were obtained from the Sichuan Agricultural University Waterfowls Breeding Farm (Ya’an, China). All experimental procedures that involved in animal manipulation were approved by the Institutional Animal Care and Use Committee (IACUC) of Sichuan Agricultural University (Chengdu, China) under Approval No. 20160067.

### 2.2. Management of Experimental Geese

For the present study, a total of 60 male Sichuan white goose with similar body weights were used as experimental material. These geese were incubated in the same batch and taken from the Waterfowl Breeding Experimental Farm of Sichuan Agricultural University (Ya’an, China). After incubating (28 d), these geese underwent the brooding stage and were raised in the same environment before the age of 120 d. Then, the experiment geese (average BW 2.75 kg) were randomly allocated into CRS and FRS (namely, 30 geese in each group) at the age of 120 d. In this study, geese in FRS were reared in an indoor area with 60 m^2^ cement playground and an 18 m^2^ fermentation bed (length (L) × width (W): 6 × 13 m). In the CRS, geese were reared in a single cage (L × W × height (H): 0.55 × 0.37 × 0.7 m, and the bottom of the cage is 1.5 m above the ground. During the experimental period, all diets (Table 1; Sanwang Agriculture and animal Husbandry Co., Ltd., Chengdu, China) and water were provided the same and *ad libitum* to geese. The temperature and humidity were kept at approximately 18 to 25 °C and at 50% and 60%, respectively. The lighting schedule was set 16 h on and 8 h off, with lights on at 08:00 A.M. for CRS and FRS.

### 2.3. Growth Performance Measurement and Sample Collection

At 270 d of age, BW and body size traits were measured individually. Body size traits including half-diving length (HDL), body straight length (TSL), body slope length (BSL), breast width (XK), breast depth (BD), fossil bone length (FBL), shank length (SL), and shank circumference (SC) were measured. Measurement of body size traits were performed in accordance with the standards of agricultural industry of the People’s Republic of China (NY/T 828-2004). After measured, blood samples (5 mL) were collected from the wing vein in the blood vessels with EDTA (Jiangsu, China) for each goose. The whole blood was placed at −4 °C for 24 h for static stratification. The serum (the light-yellow fluid) was collected and stored at −20 °C for further determination of geese immune status.

Then, 18 geese were randomly selected from CRS and FRS, respectively, for slaughter and tissue collection. After slaughter (carbon dioxide to render the geese unconscious, and then cervical dislocation to euthanize the geese), three tissues were collected, namely, spleen, thymus, and bursa of Fabricius. Subsequently, quick measurements of the spleen, thymus, and bursa of Fabricius weight were performed, before calculating the immune organs index using the following formula: organ index = organ weight (g)/body weight (g) × 100%. Lastly, approximately 1 g of spleen tissue was collected and placed in liquid nitrogen, before storing at −80 °C until RNA extraction.

### 2.4. Determination of Immune Organ Parameters and Serum Immune Cytokine Profiles

In this experiment, the immune status of geese under CRS and FRS was mainly explored through two aspects: immune organ development and serum immunity cytokines levels. In terms of immune organs, we calculated six immune organ parameters, including organ weight (spleen, thymus, and bursa of Fabricius weight) and organ index (spleen, thymus, and bursa of Fabricius index). In addition, we measured the levels of immune globulin-G (IgG), interleukin-1β (IL-1β), interleukin-4 (IL-4), interleukin-6 (IL-6), and interferon-γ (IFN-γ) in serum using ELISA kits (Shanghai, China), respectively.

### 2.5. Histological Observation

Firstly, the spleen tissue was fixed with 4% paraformaldehyde for 24 h, ensuring that paraformaldehyde completely immersed the spleen tissue. Then, the fixed spleen tissue was dehydrated completely using 75%, 85%, 95%, and absolute alcohol. After the dehydration of the spleen tissue, it was swept in xylene, unfiltered, and embedded in melted paraffin. Then, the embedded sample was cut using a rotary microtome (Leica RM2235, Oskar-Barnack, Munich, Germany), with a slice of 5 um thickness. During staining, HE staining was performed on the cross-sectional area of the sliced sample and photographed using a digital three camera microscope BA410-Digital (Motic China Group Co. Ltd., Xiamen, China). Lastly, the red pulp (RP), central artery (CA), splenic trabecula (TL), splenic corpuscle (AL), and trabecular artery (TA) were counted by Image-Pro Plus 6.0.

### 2.6. RNA Isolation and Sequencing

In this experiment, we used RNeasy Mini Kit (QIAGEN, Beijing, China) to extract total RNA from the spleen tissues of CRS (*n* = 3) and FRS (*n* = 3) geese. After extraction, the integrity of RNA from different samples was detected using Agilent Bioanalyzer 2100 (Agilent, CA, USA). Then, we selected high-integrity (RIN value from 7.9 to 9.7) spleen tissues to construct RNA libraries by Glibizzia Bioscience (Glibizzia, Beijing, China). All RNA-libraries were sequenced by the DNBSEQ-T7-PE150 (HuaDa, Shenzhen, Chian). Standard quality control of FastaQC software was used to filter low-quality reads and obtain clean reads.

### 2.7. Transcriptome Alignment and Assembly

HISAT2 (version 2.1.0) software was used to compare clean reads with the reference genome of Sichuan white goose (data being published) that we assembled [11]. In terms of file conversion, SAMtools (version 1.13) software was used to convert the output SAM files into BAM files and sort them [12]. Subsequently, featureCounts (version 2.02) software to calculate the counts values of each transcript, using the following calculation criteria: the gene length and the read count were mapped to the transcript [13].

### 2.8. Identification of the DEGs and Functional Analysis

According to the above groups (CRS and FRS), the DESeq2 package was used to identify differentially expressed genes (DEGs), and p adj values < 0.05 and |log_2_FC| > 1 as the criteria [14]. KOBAS 3.0 was used to functional analysis of DEGs [15]. The STRING 10 database (http://string-db.org/ accessed on 19 Augst 2022) was used to analyze DEGs–protein interaction networks in this study. Finally, the Cytoscape (version 3.2.1) software was used to network visualization [16].

### 2.9. Statistics Analysis

The experimental data were analyzed using the SPSS 23.0 software (IBM, Armonk, NY, USA). Differences in growth performance traits, weight for spleen, thymus, bursa of Fabricius, spleen histomorphological characteristics, and immune cytokines profiles between CRS and FRS through ANOVA testing. Furthermore, we analyzed the significance of different traits between CRS and FRS through *t*-tests; when *p* < 0.05, a statistically significant difference was recognized.

## 3. Results

### 3.1. Effects of CRS vs. FRS on Goose Growth Performance

The results of growth performance are shown in Table 2. Geese in FRS showed higher BW (*p* < 0.05). As for body size traits, compared with the CRS, the BSL, TSL, FBL, XK, and SC in FRS were significantly higher (*p* < 0.05).

### 3.2. Effects of CRS vs. FRS on Goose Immune Organs Development and Serum Immune Cytokine Profiles

The results of immune organ weights and indices was shown in Table 3. As for immune organs, the weight and organ index of the spleen were significantly higher in geese under CRS (*p* < 0.05). Noticeably, the bursa of Fabricius of geese under FRS was degenerated, while that under CRS was retained. As shown in Figure 1, at the serum level, the immune IgG and IL-6 levels were significantly higher in geese under CRS (*p* < 0.05), while the IL-4, IL-1β, and IFN-γ levels were not significantly difference between the CRS and FRS groups.

### 3.3. Effects of CRS vs. FRS on Goose Spleen Histomorphology

To determine the changes in spleen between CRS and FRS, histological analysis was performed (Figure 2A). As shown in Figure 2B, compared with the FRS, the diameter of splenic corpuscle (AL), the area of spleen trabecula (TL), and red pulp (RP) were significantly higher in geese under CRS (*p* < 0.05).

### 3.4. Overview of the mRNA Transcriptome of Spleen between CRS and FRS

As shown in Appendix A, total of 51.8 Gb raw reads were obtained through RNA-seq, about 8.6 Gb clean reads of each sample were obtained through quality control, and the Q20 and Q30 ranges were 97.68–98.03% and 92.82–93.76%, respectively. Compared with the reference genome, it was shown that the mapping rate was between 91.31 and 92.71%.

### 3.5. Identification of the Spleen DEGs between CRS and FRS

In this study, PCA results indicated that the clustering of samples within the same group was relatively concentrated, while the clustering of samples among groups was relatively dispersed (Figure 3A). Furthermore, there were 251 DEGs (42 downregulated, 209 upregulated) identified in the spleen (Figure 3B and Appendix A). The clustering heatmap results also showed that the gene expression patterns had significant differences among CRS and FRS goose spleen (Figure 3C).

### 3.6. Functional Analysis of the DEGs Identified between CRS and FRS

To better understand the possible functions of DEGs in immunity, the spleen DEGs between CRS and FRS were annotated with the GO database. We found that 251 DEGs identified in the spleen were mainly enriched in the biological process (BP), cellular component (CC), and molecular function (MF) categories of GO classification (Appendix A). The top 30 of 229 GO terms were obtained and are presented in Figure 4A. Most DEGs in the spleen were enriched in inflammatory response, positive regulation of transcription by RNA polymerase II, scavenger receptor activity, angiogenesis, extracellular space, positive regulation of kinase activity, activation of protein kinase B activity, immune response, positive regulation of protein kinase B signaling, and positive regulation of tyrosine phosphorylation of STAT protein (Figure 4B).

According to the KEGG pathway enrichment results, 251 DEGs were enriched in 78 pathways (Figure 4C and Appendix A). The DEGs were found significantly enriched in phenylalanine metabolism, phenylalanine, tyrosine, and tryptophan biosynthesis, neuroactive ligand–receptor interaction, ECM receptor interaction, calcium signaling pathway, regulation of actin cytoskeleton, and MAPK signaling pathway (*p* < 0.05).

### 3.7. Network Analysis of the DEGs Involved in Regulating Immune Status between CRS and FRS

In order to further explore the interaction relationship between candidate DEGs, we conducted a more comprehensive bioinformatics analysis of the DEGs using PPI networks (Figure 5A). Functional analysis results showed that the pathways of “phenylalanine metabolism” and “phenylalanine, tyrosine, and tryptophan biosynthesis” were significantly enriched in this PPI network. The top highest degree genes including *VEGFA*, *FGF2*, *NGF*, *GPC1*, *NKX2-5*, *FGFR1*, *FGF1*, *MEIS1*, *CD36*, and *PAH*. Notably, our results showed that geese in CRS could enhanced their immune ability through “phenylalanine metabolism” pathway to respond to virus (Figure 5B).

## 4. Discussion

It is generally believed that growth performance is the most direct indicator for evaluating poultry production efficiency, and a study found that it is influenced by the rearing system [17]. In a study by Chen et al. [18], the ducks reared in the intensive system compared with those of semi-intensive system ducks, showed that the growth performance and meat quality were significantly different. Similarly, in Chaohu ducks and Moulard ducks, NRS had better growth performance compared to FRS, as reflected in its higher final BW and ADG [7,19]. In geese, different rearing systems showed a strong association with growth performance, carcass traits, and meat quality [20,21]. Furthermore, a study on Yangzhou geese showed that geese reared in NRS had higher final BW and ADG than those in FRS [22]. In this study, geese in the FRS showed higher BW and body size traits (BSL, TSL, FBL, XK, and SC), illustrating that the growth performance of geese under FRS was better.

In terms of immune status, a previous study reported that ducks reared in NRS improved their immune function by promoting the development of the thymus and spleen when compared with FRS [10]. Consistent with this, we found that the weight and organ index of spleen were significantly higher in geese under CRS, indicating that the development of spleen in CRS geese was promoted. Numerous studies have shown that the bursa of Fabricius is a key humoral immune organ in poultry and plays an important role in the differentiation of B lymphocytes [23,24]. Interestingly, the bursa of Fabricius of geese under FRS were degenerated at 270 d old, while those under CRS were retained, implying that the immune function was enhanced in CRS geese. Furthermore, this study also showed that the serum immune cytokines were extensively affected by rearing systems, with higher IgG and IL-6 levels in geese under CRS. In general, the higher interleukin and IgG levels in CRS geese indicated a better immune status. Previous studies have shown that interleukin contribute to T helper type-2 cell activity [25,26,27], and IFN-γ contributes to T helper type-1 cell activity [28]. Taken together, our results suggested that CRS could promote immune organs development and increase the levels of certain immune cytokines to improve immune ability of geese.

To further understand the molecular mechanisms underlying how the two rearing systems differentially affected goose immune functions, we used RNA-seq to compare the spleen transcriptomic mRNA profiles between CRS and FRS. As a result, a total of 251 DEGs were identified in the spleen, and most of them were enriched in the immune response-related GO terms, including the positive regulation of transcription by RNAPII (RNA polymerase II), kinase activity, scavenger receptor activity, inflammatory response, activation of protein kinase B activity, immune response, STAT protein tyrosine phosphorylation, and protein kinase B signaling, and KEGG results showed that DEGs was significantly enriched in the pathways of phenylalanine metabolism, neuroactive ligand–receptor interaction, calcium signaling, ECM receptor interaction, phenylalanine, tyrosine and tryptophan biosynthesis, regulation of actin cytoskeleton, and MAPK signaling, which were crucial for regulating immune functions. Previous studies have shown that the phenylalanine metabolism, MAPK signaling, and regulation of actin cytoskeleton pathways plays a crucial in regulating poultry immune ability [23,29,30].

Furthermore, we performed a network analysis to reveal the role of these identified DEGs and signaling pathways in regulating goose immune function. It was observed that the “phenylalanine metabolism” pathway could play a critical role in affecting geese immune ability between different rearing systems. In this study, DEGs identified in the spleen between CRS and FRS significantly enriched the phenylalanine metabolic pathway. Moreover, transcriptome studies in human [31], chicken [32], fish [33], and white shrimp Litopenaeus vannamei [34] have also shown that the important role of phenylalanine metabolic pathway in the control of immune ability. Moreover, our results showed that all the DEGs (PAH, TAT, and AOC3) enriched in the phenylalanine metabolism pathway were significantly upregulated, indicating that the phenylalanine metabolism pathway was promoted in CRS geese. Notably, the expression of PAH (log2FC = 5.96) was significantly different in the spleen between CRS and FRS. A previous study found that, as an important metabolic enzyme, PAH synthesizes catecholamines and melanin by providing the starting material; hence, it is involved in immune defense reactions [35]. The above results suggested that overactivation of the phenylalanine metabolism pathway in the spleen of CRS geese may be one of the reasons for the enhanced immune ability.

## 5. Conclusions

In conclusion, compared to CRS, the growth performance was improved in geese under FRS. CRS increased levels of IL-6 and IgG in serum and promoted the development of the bursa of Fabricius and spleen to improve goose immune function. Furthermore, we constructed the first transcriptomic profiles of the spleen between CRS and FRS. Notably, the expression of PAH was significantly different in the spleen between CRS and FRS, indicating that it may be a candidate gene related to goose immune status. Moreover, the phenylalanine metabolism pathway was promoted in the spleen, which exerted a positive effect on immune ability in geese under CRS. These results can provide a reliable reference for selecting a dryland rearing system in geese.

## Figures and Tables

**Figure 1 animals-13-02682-f001:**
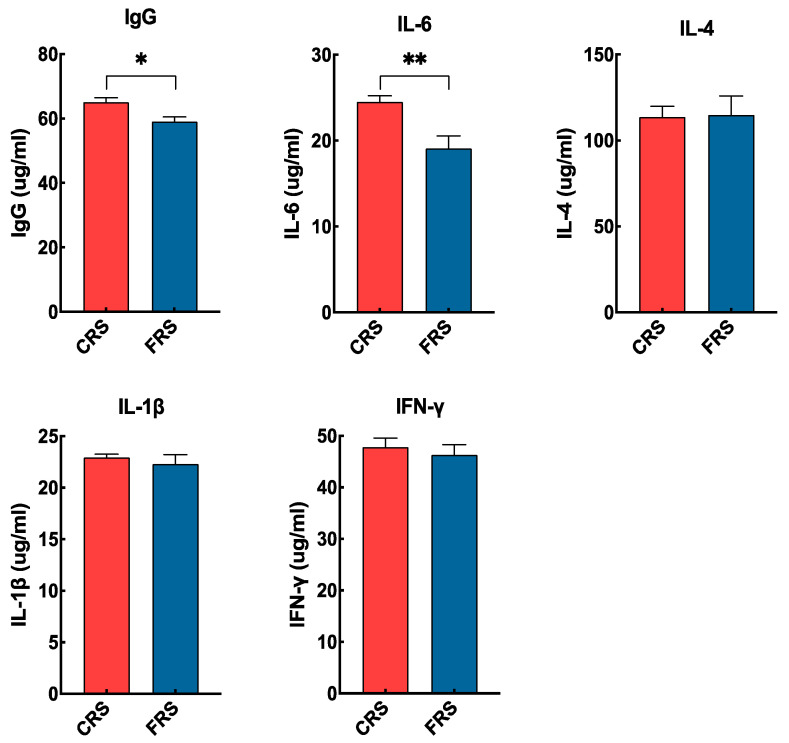
Effects of CRS and FRS on serum immune cytokine profiles of geese. Data are displayed as the means ± SEM. For both CRS and FRS, *n* = 7. * and ** indicate a significant difference at *p* < 0.05 and *p* < 0.01 between the two different groups, respectively. Abbreviations: IL, interleukin; IFN-γ, interferon-γ; Ig, immune globulin; CRS, cage rearing system; FRS, floor rearing system.

**Figure 2 animals-13-02682-f002:**
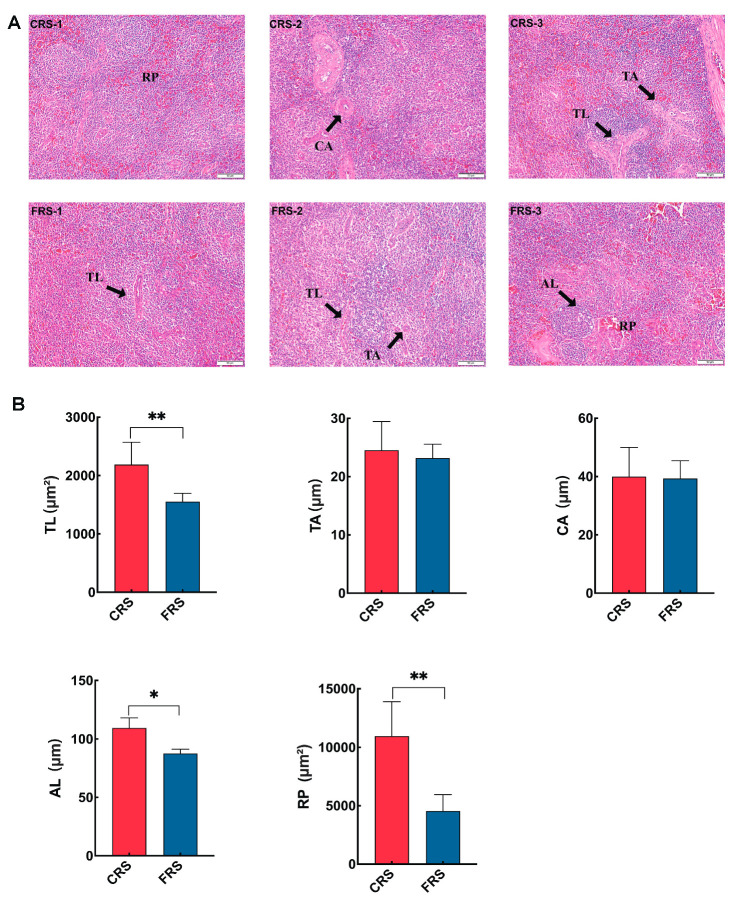
Histomorphological difference of the spleen between CRS and FRS. (**A**) Morphological differences of spleen between CRS and FRS (200×). (**B**) Spleen histomorphological parameters of geese between CRS and FRS. Abbreviations: RP, red pulp; CA, central artery; TL, splenic trabecula; AL, splenic corpuscle; TA, trabecular artery. * and ** indicate a significant difference at *p* < 0.05 and *p* < 0.01 between the two different groups, respectively.

**Figure 3 animals-13-02682-f003:**
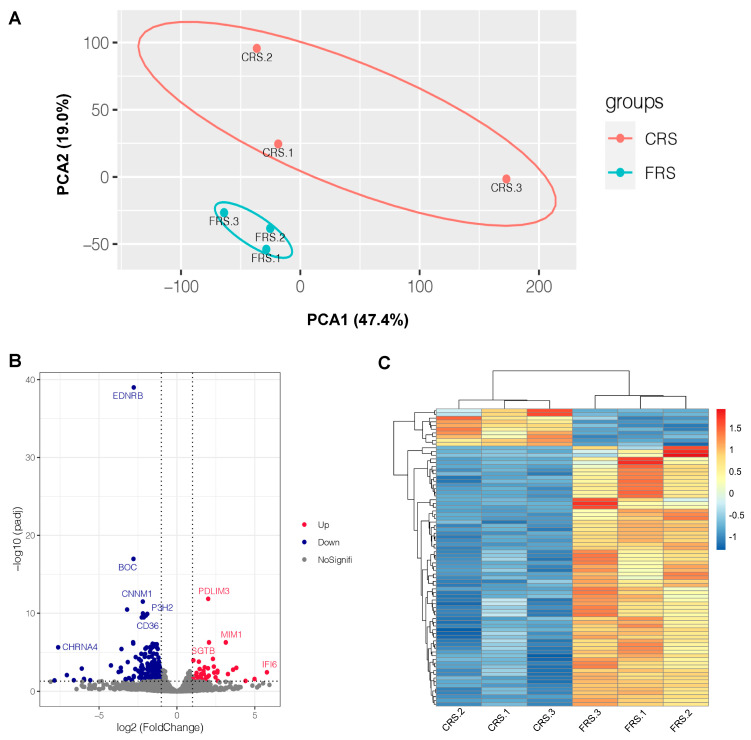
Principal component analysis and DEGs analysis of the spleen between CRS and FRS. (**A**) Principal component analysis. PCA1, principal component 1; PCA2, principal component 2. (**B**) Volcano diagram of DEGs of the spleen between CRS and FRS. The red and blue dots represent upregulated genes with significant differential expression and downregulated genes with significant differential expression, respectively. (**C**) Hierarchical clustering of DEGs between CRS and FRS.

**Figure 4 animals-13-02682-f004:**
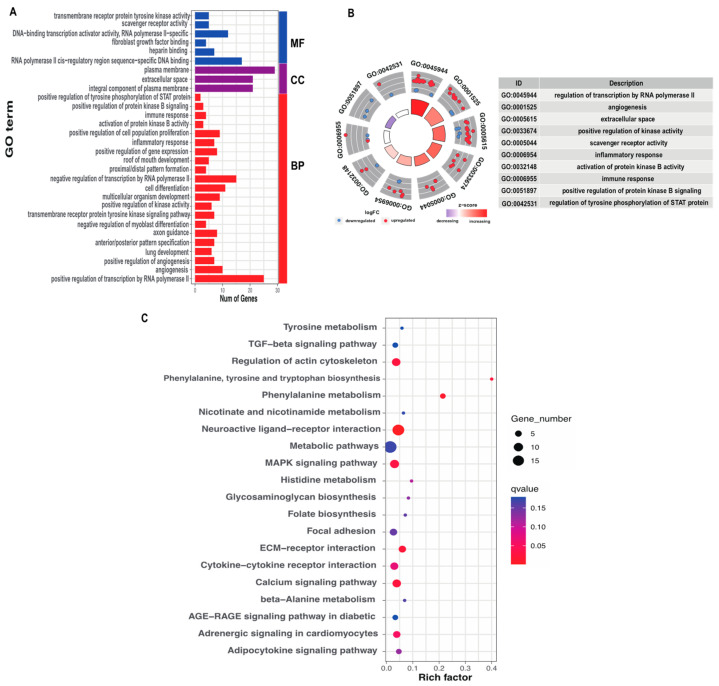
Functional analysis of the DEGs between CRS and FRS. (**A**,**B**) GO terms enriched by DEGs. (**C**) Top 20 significantly enriched KEGG pathways. The Rich factor represents the degree of enrichment: the larger its value, the higher the degree of enrichment, calculated using the following equation: Rich factor = the number of DEGs in the pathway/the total number of genes in the pathway. The q-value represents the *p*-value corrected for multiple hypothesis tests, with a range of 0–1: the smaller the q-value, the more significant the enrichment.

**Figure 5 animals-13-02682-f005:**
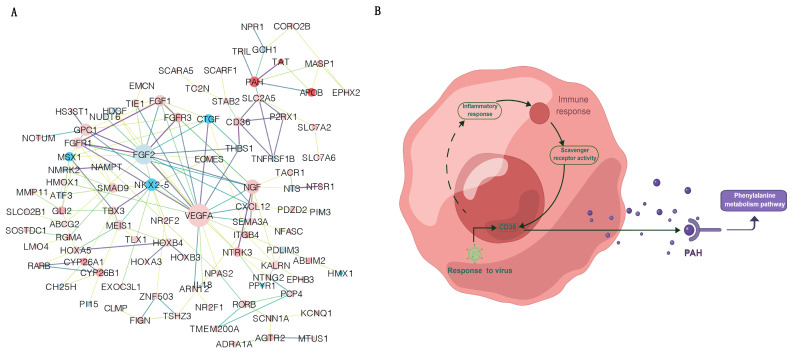
Network analyses of the DEGs between CRS and FRS. (**A**) PPI networks of the DEGs. The network was constructed by 251 DEGs based on DEGs, GO enrichment, and KEGG pathway, which consisted of 2 significantly enriched pathways. (**B**) Regulation network construction involved in immune function changes of geese under CRS and FRS.

**Table 1 animals-13-02682-t001:** Ingredients and nutrients composition of basal diets in CRS and FRS geese.

Items	Stage (28–270 d)
Ingredients	
Corn (%)	57.70
Soybean meal (%)	27.50
Wheat middling (%)	7.50
Wheat bran (%)	2.00
Calcium hydrogen phosphate (%)	1.62
Soybean oil (%)	1.40
Limestone powder (%)	0.93
NaCl (%)	0.35
Vitamin and mineral premix (%)	1.00
Total (%)	100
Nutrients	
Metabolizable energy (Mcal/kg)	2900
(Mcal/kg) Dry matter (%)	87.12
Crude protein (%)	17.50
Crude fat (%)	4.13
Crude fiber (%)	3.00
Calcium (%)	0.85
Total phosphorus (%)	0.65
Available phosphorus (%)	0.40
Lysine (%)	0.85
Methionine (%)	0.40
Methionine + cystine (%)	0.70
Threonine (%)	0.60
Tryptophan (%)	0.19

**Table 2 animals-13-02682-t002:** Effects of FRS and CRS on growth performance of geese.

Items	CRS (*n* = 18)	FRS (*n* = 18)	*p*-Value
BW (g)	4195.6 ± 541.5	4603.9 ± 327.2	0.01
HDL (cm)	79.2 ± 2.8	79.6 ± 2.2	0.60
BSL (cm)	31.0 ± 1.6	32.2 ± 1.5	0.01
TSL (cm)	29.8 ± 1.6	30.8 ± 1.4	0.03
FBL (cm)	16.4 ± 0.9	17.1 ± 0.8	0.01
BD (cm)	9.5 ± 0.9	10.0 ± 1.9	0.31
XK (cm)	11.7 ± 0.8	12.9 ± 0.8	0.01
SL (cm)	9.7 ± 0.4	10.2 ± 1.7	0.17
SC (cm)	5.3 ± 0.3	5.7 ± 0.4	0.01

Abbreviations: CRS, cage rearing system; FRS, floor rearing system; HDL, half-diving length; BSL, body slope length; TSL, body straight length; FBL, fossil bone length; XK, breast width; BD, breast depth; SL, shank length; SC, shank circumference. All results are presented as the mean ± standard deviation (SD).

**Table 3 animals-13-02682-t003:** Effects of FRS and CRS on immune organ development of geese.

Items	CRS (*n* = 18)	FRS (*n* = 18)	*p*-Value
Spleen weight (g)	2.41 ± 0.71	1.77 ± 0.69	0.017
Spleen index (‰)	0.65 ± 0.37	0.36 ± 0.12	
Thymus weight (g)	1.98 ± 0.60	1.78 ± 0.76	0.468
Thymus index (‰)	0.47 ± 0.13	0.38 ± 0.17	

Abbreviations: CRS, cage rearing system; FRS, floor rearing system. All results are presented as the mean ± standard deviation (SD).

## Data Availability

Data will be available upon request to the corresponding author.

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
