# Peer review of "Effects and Mechanisms of Cage versus Floor Rearing System on Goose Growth Performance and Immune Status"

_animals, 2023, doi:10.3390/ani13162682_

Round 1
Reviewer 1 Report
It is a interesting and important study.
Important information is missing in the Material and Methods for the analysis of this study. It is not clear in the Material and methods item why the experiment started at 120 days and ended at 270 days of age.
There is a lack of information about the diet and light regime offered to the birds, which are essential to assess the veracity of the differences found between the experimental groups. Likewise, information on the initial weight of the birds and how they were distributed among the experimental groups at 120 days of age. Data on feed intake, water intake, and weight gain between 120-270 days are also essential for data analysis.
How were the experimental repetitions performed? In the CR system, were there 30 ages with one bird each? That is, 30 repetitions of 1 bird? Each bird a unit? What about the FR system? Were the 30 birds housed together? So there was no replication? It is important that authors seek expertise in statistics, who will have greater knowledge about the adequacy or otherwise of the experimental design to support the data.
Author Response
We really appreciate all your comments and suggestions. Thanks very much for taking your time to review this manuscript. Those comments are all valuable and very helpful for revising and improving our paper. We have revised the manuscript accordingly, and please see the attachment.

Reviewer 2 Report
The study is about the cultivation system, which has an important place in the last period. Therefore, the reader effect is high. Abstract and introduction are sufficient and good.
The material and method section does not adequately explain the study. The method is presented according to the findings obtained. The literature is sufficient.But; It would be great if the consumed feed content is given. In addition, the trial period (light-dark) day length should be given. It should be noted that it is the summer period or the winter period. As descriptive information, the weight of the geese in this period (120 d.) should be given. How do we know the difference between the last period weights? Maybe it was due to the initial live weight.
Results section is sufficient and understandable.
More geese-related literature should be used in the discussion section.
The conclusion is sufficient.
Although the article is good in general, it should be revised taking into account the suggestions in the text. Otherwise it will be missing.

Author Response

(The authors gave the same response as above.)

Reviewer 3 Report
The objectives of the current study were to compare the effects of cage-rearing system (CRS) and floor-rearing system (FRS) on goose growth performance and immune status and also compare the spleen transcriptome profiles between CRS and FRS using RNA-seq. The Introduction chapter provides an overview of the world's knowledge on this subject. The material used in the research is sufficiently numerous, and the research methods used are correct. Description of environmental and nutritional conditions is required in Materials and methods section. Description results need improvement in a few places. The discussion is exhaustive. Summary of the results are correct. The proposed changes are listed below.
General comments:
Please prepare the article in accordance with the instructions for authors:
L7-12 In parentheses, give initials of first and last names instead of full name
For significance please use lowercase "p" in italic instead of uppercase "P" throughout the main article
Add „a dot” after the table title
When quoting two references, do not use spaces, for example [22,23] instead of [22, 23]
The references chapter change are nedded: volume in italic, year in bold, for page ranges use long (-) from the symbol function, instead of short (-) from the keyboard
In References chapter please use a "dot" after each abbreviation, for example Poult. Sci. instead of Poult Sci
Detailed Comments:
L30 IgG or IgY?
L49+ may add something that in many countries of the world the cage-rearing system in geese is unknown. Geese are reared in an extensive or semi-intensive system (with access to a paddock and water (swimming pools, ponds, lakes)
L61 final body weight - how many days/weeks was the rearing period for ducks?
L53 which indoor system is the most popular in China?
L55 Please write something about the research results of Li et al. (2022), Poult. Sci. 101, 101931
L79 and others, 2.1., dot after the second number
L92 What was the type of floor in the cage-system, mesh?, slatted?
L92 what building, closed without windows?, what temperature, humidity, lighting program? What nutrition program, level of CP and ME, Ca, P, Liz.?
L114 How indexes for immune organs were calculated?
L160 What statistical characteristics are presented in the article?, or normal distribution of features?
L181 What about thymus?
L183 What about the description of other attributes in Figure 1
L197 add abbreviations (TL or AL or RP) in parentheses
L267 Add something about the research results of
Sabek et al. 2016 in Asian J. Poult. Sci. 10, 153-157
Liu et al. Poult. Sci. 90, 653-659;
Chen et al. 2022 Animal Biotechnology doi: 10.1080/… Effect of different free-range system on…
L269 [19,20] instead of current form
L280 [22,23] instead of current form
Author Response

(The authors gave the same response as above.)

Round 2
Reviewer 2 Report
The authors have taken into account the necessary recommendations. Necessary dedication has been provided to improve the article. The introduction provides sufficient background and contains all relevant references. Citations are sufficient and relevant to the article. The research design is suitable for the article. The method of the study is adequately described. The results include the findings and are clearly legitimate. The discussion section and conclusions describe the findings.